# Diagnostic Performances of PET/CT Using Fibroblast Activation Protein Inhibitors in Patients with Primary and Metastatic Liver Tumors: A Comprehensive Literature Review

**DOI:** 10.3390/ijms25137197

**Published:** 2024-06-29

**Authors:** Federica Manuppella, Giusi Pisano, Silvia Taralli, Carmelo Caldarella, Maria Lucia Calcagni

**Affiliations:** 1Dipartimento Di Diagnostica Per Immagini e Radioterapia Oncologica, UOC Di Medicina Nucleare, Fondazione Policlinico Universitario Agostino Gemelli IRCCS, Largo Agostino Gemelli, 8, 00168 Rome, Italy; federica.manuppella01@icatt.it (F.M.); giusi.pisano01@icatt.it (G.P.); silvia.taralli@policlinicogemelli.it (S.T.); marialucia.calcagni@unicatt.it (M.L.C.); 2Dipartimento Universitario Di Scienze Radiologiche Ed Ematologiche, Università Cattolica del Sacro Cuore, Largo Francesco Vito, 1, 00168 Rome, Italy

**Keywords:** liver primary tumor, liver metastases, radiolabeled FAPIs, performance, tumor-to-background ratio, comparison

## Abstract

PET/CT using radiolabeled fibroblast activation protein inhibitors (FAPIs) is a promising diagnostic tool in oncology, especially when non-increased and/or physiologically high [^18^F]FDG uptake (as in liver parenchyma) is observed. We aimed to review the role of PET/CT using radiolabeled FAPIs in primary and/or metastatic liver lesions, and to compare their performances with more “conventional” radiopharmaceuticals. A search algorithm based on the terms “FAPI” AND (“hepatic” OR “liver”) was applied, with the last update on 1st January 2024. Out of 177 articles retrieved, 76 studies reporting on the diagnostic application of radiolabeled FAPI PET/CT in at least one patient harboring primary or metastatic liver lesion(s) were fully analyzed. Although there was some heterogeneity in clinical conditions and/or study methodology, PET/CT with radiolabeled FAPIs showed an excellent performance in common primary liver malignancies (hepatocarcinoma, intrahepatic cholangiocarcinoma) and liver metastases (mostly from the gastrointestinal tract and lungs). A higher tumor-to-background ratio for FAPIs than for [^18^F]FDG was found in primary and metastatic liver lesions, due to lower background activity. Despite limited clinical evidence, radiolabeled FAPIs may be used to assess the suitability and effectiveness of FAPI-derived therapeutic agents such as [^177^Lu]Lu-FAPI. However, future prospective research on a wider population is needed to confirm the excellent performance.

## 1. Introduction

Positron emission tomography/computed tomography (PET/CT) is a nuclear medicine imaging technique increasingly used in oncology for early diagnosis and treatment evaluation in several neoplasms, especially in the absence of morphological abnormalities. Although the glucose metabolism-directed fluorine-18 fluorodeoxyglucose ([^18^F]FDG) is the mainstay of PET/CT in most applications, the availability of new, more target-specific radiopharmaceuticals evaluating different metabolic pathways and/or receptor expression has broadened the possible clinical horizons of PET/CT [1,2]. Some neoplasms (e.g., prostate, renal cell carcinoma, hepatocarcinoma, or G1 neuroendocrine tumors) are not [^18^F]FDG-avid; moreover, neoplasms growing in sites with physiologically high [^18^F]FDG uptake (brain, oro-pharyngeal mucosa, liver, bowel, and urinary tract) may be missed.

The prevalent component (about 90%) of a tumor is the tumor microenvironment (TME), or stroma, composed of extra-cellular matrix, cancer-associated fibroblasts (CAFs), and immune, precursor, mesenchymal stromal, and endothelial cells exhibiting pro-tumoral or tumor-suppressive activity [3]. Fibroblast activation protein (FAP), a membrane-anchored serine protease, is highly over-expressed on CAFs membrane, therefore representing a marker of CAFs activation [4]; conversely, in healthy tissues, FAP is rare (quiescent fibroblasts mostly express dipeptidyl peptidase 4). Since FAP over-expression is associated with progression, metastatic spread, and treatment resistance in tumors, new PET-suitable radiopharmaceuticals based on FAP-specific inhibitors (FAPIs) have been developed in recent years [5,6,7]. Since the first studies in 2018–2019 [7,8,9], the intense uptake of gallium-68-labeled FAPIs, with a high tumor-to-background ratio (TBR), has been demonstrated in sarcomas, esophageal, breast, lung, and pancreatic cancers, and cholangiocarcinoma [5,6]. After labeling with alpha- or beta-emitting radioisotopes (e.g., lutetium-177, yttrium-90, actinium-225), FAP-targeting ligands may be used as a new anti-cancer radioligand therapy (RLT) [10,11,12,13,14,15,16].

The demonstration of molecular crosstalk between CAFs and malignant cells, leading to the secretion of growth factors and cytokines (SDF-1, HGF, FGF, IL-6, TGF-β, EGF) that promote proliferation and neo-angiogenesis [17,18,19], is the basis for using PET/CT with FAPI radiopharmaceuticals in liver malignancies. Recent clinical studies on diagnostic and/or theranostic applications of FAPI radiopharmaceuticals in oncology mostly include patients with mixed malignancies, in heterogeneous clinical settings; in patients with suspected malignant liver disease, either primary like hepatocellular carcinoma (HCC) and intrahepatic cholangiocarcinoma (ICC), or liver metastases from various cancers, there is some evidence regarding the high diagnostic performance of FAPI radiopharmaceuticals, with it being comparable to MRI and better than [^18^F]FDG, especially in the early stages [20,21,22]. Furthermore, FAPI radiopharmaceuticals may outperform [^18^F]FDG in detecting extrahepatic involvement from HCC or ICC [22], and they are able to differentiate ICC (with a higher TBR) from HCC and liver metastases (with a lower TBR), despite them having a limited role in predicting HCC grade [21]. In patients with extrahepatic malignancies, there is initial evidence of a higher value of FAPI radiopharmaceuticals than [^18^F]FDG in detecting liver metastases, due to a higher uptake and/or TBR [23,24].

However, to the best of our knowledge, a review exclusively focused on the diagnostic role of PET using FAPI radiopharmaceuticals in patients with primary and/or metastatic liver lesions is still lacking. Therefore, we aimed to review the available literature on this topic to provide a more comprehensive insight into the role of PET using FAPI radiopharmaceuticals in the management of patients with liver lesions.

## 2. Results

### 2.1. Literature Research

The initial literature search retrieved 177 articles and, after applying the above-reported exclusion and inclusion criteria, 76 papers dealing with the diagnostic application of radiolabeled FAPI PET in at least one patient with neoplastic liver lesions (primary tumor or metastasis) were included [Figure 1]: 39 evaluated FAPI PET diagnostic imaging exclusively in patients with liver metastases from mixed primary malignancies [23,24,25,26,27,28,29,30,31,32,33,34,35,36,37,38,39,40,41,42,43,44,45,46,47,48,49,50,51,52,53,54,55,56,57,58,59,60,61]; 23 evaluated FAPI PET diagnostic imaging exclusively in patients with primary liver tumors [20,22,62,63,64,65,66,67,68,69,70,71,72,73,74,75,76,77,78,79,80,81,82]; 14 evaluated FAPI PET diagnostic imaging in patients with both primary liver tumors and metastatic liver involvement [21,83,84,85,86,87,88,89,90,91,92,93,94,95].

### 2.2. Clinical and Methodological Studies’ Characteristics

Out of 76 studies analyzed, 45 [20,21,25,27,28,29,30,32,35,36,37,38,40,43,44,49,51,53,54,56,57,64,66,68,69,70,74,76,77,78,79,80,81,83,84,85,86,87,90,91,92,93,94,95] were prospective and 22 [22,23,24,26,31,34,39,41,42,45,47,48,55,58,59,60,61,62,65,67,88,89] were retrospective, while in 9 studies, the design was not specified [33,46,50,52,63,71,72,73,75]. Regarding study populations, 28 studies [28,30,32,33,35,36,38,39,44,46,49,50,51,52,54,63,64,66,68,69,70,71,72,73,74,75,76,77] included only 1 patient with malignant liver lesions, 19 [24,25,27,36,37,41,43,45,48,55,56,57,59,65,79,83,84,85,92] included 2–14 patients with malignant liver lesions, and 19 studies [20,21,22,23,58,61,62,67,78,80,81,82,86,88,89,90,93,94,95] evaluated a larger population with liver malignancies (ranging from 16 to 67 patients), while in 10 studies [26,29,31,34,42,47,53,60,87,91], the number was not clearly reported. Concerning the 23 studies focused on the evaluation of primary liver tumors (whose main clinical and methodological characteristics are reported in Table 1) [20,22,62,63,64,65,66,67,68,69,70,71,72,73,74,75,76,77,78,79,80,81,82], 5 included both HCC and ICC [21,22,62,65,81], 7 exclusively HCC [67,69,70,71,72,78,82]; 4 studies included only ICC [63,75,79,80], while 6 case reports evaluated extranodal marginal zone lymphoma [64], low-differentiated neuroendocrine carcinoma [66], primary liver inflammatory myofibroblastoma [73,77], hepatic adenocarcinoma [74], and perivascular epithelioid cell tumor (PEComa) [76], respectively. One case report did not specify the histotype, having evaluated features of a tumoral thrombus from a hepatic lesion [68]. In the 39 papers (whose main clinical and methodological characteristics are reported in Table 2) focused on liver metastases [23,24,25,26,27,28,29,30,31,32,33,34,35,36,37,38,39,40,41,42,43,44,45,46,47,48,49,50,51,52,53,54,55,56,57,58,59,60,61], the primitive neoplasm was mostly located in the colorectal, gastric, and pancreatic–biliary districts, while in 5 studies, the metastases were derived from NETs (15%) [24,35,39,40,56].

FAPI diagnostic performances in detecting liver malignancies were compared with [^18^F]FDG [21,22,23,24,25,26,27,29,30,32,34,35,36,37,38,39,41,42,43,44,45,46,47,48,49,50,52,53,54,57,58,59,60,61,62,63,64,66,67,68,69,71,72,74,76,77,78,79,80,81,82,92,93], [^68^Ga]Ga-DOTATATE, or [^68^Ga]Ga-DOTATOC [24,28,33,35,40,56,89], [^131^I]mIBG [28], or [^11^C]acetate [39]; three studies made a comparison among different FAPI tracers [84,91,92]. In four studies, FAPIs were used for RLT suitability assessment [28,31,40,56].

Concerning diagnostic modalities, PET/CT was the most used, but Positron Emission Tomography/Magnetic Resonance (PET/MR) was performed in three [27,37,68], or dedicated liver PET/MRI after PET/CT in one study [62]. The most frequently administered radiopharmaceutical was [^68^Ga]Ga-FAPI-04, except for [^18^F]FAPI-42 in two studies [47,84], Al^18^F-NOTA-FAPI-04 in two studies [30,69], [^68^Ga]Ga-LNC1007 + [^68^Ga]Ga-FAPI-02 in one study [92], [^68^Ga]Ga-FAPI-46 in two studies [55,79], [^68^Ga]Ga-DOTA.SA.FAPI in two studies [58,61], [^18^F]FAPI-74 in one study [93], [^18^F]FAPI-04 in a case report [54], and [^18^F]FAPI-42 + [^68^Ga]Ga-FAPI-04 in one study [59]; in three other papers, a not otherwise specified ^18^F-labeled FAPI radiopharmaceutical was employed [71,78,90].

The time interval from the tracer injection to the PET/CT image acquisition was usually 60 min, with a field of view from the vertex to the upper or mid-thighs. The injected activity was heterogeneous, from 1.8 to 3.7 MBq/Kg. In two studies performing dynamic acquisition, the scan duration ranged from 20 to 60 min [84,88]. In one study, dual timepoint acquisition (at 20 and 60 min post-injection) was performed [55].

When semi-quantitative analysis was performed, maximum and mean Standardized Uptake Values (SUVmax and SUVmean) were used, as well as tumor-to-background ratio (TBR), with the normal background liver selected as a reference.

### 2.3. PET Imaging

#### 2.3.1. Physiological Radiolabeled FAPI Liver Uptake

Preliminary studies already reported a significantly lower liver physiological [^68^Ga]Ga-FAPI background uptake when compared with [^18^F]FDG PET/CT (average SUVmax 1.69 vs. 2.77) [5], as confirmed more recently with 71 patients (normal liver mean SUVmax 1.42 vs. 3.10 with [^68^Ga]Ga-FAPI and [^18^F]FDG, respectively) in another study [34].

Divergent findings are available when comparing background liver uptake according to the presence of liver cirrhosis: Guo et al. [22] reported a higher [^68^Ga]Ga-FAPI-04 liver SUVmax in patients with cirrhosis (4.84 ± 1.64 vs. 1.99 ± 0.65; *p* < 0.01) but no differences in [^18^F]FDG (3.3 ± 0.68 vs. 3.21 ± 0.39, *p* = 0.56), and a higher [^68^Ga]Ga-FAPI TBR in non-cirrhotic patients. In another study [67], only [^68^Ga]Ga-FAPI-04 SUVmean (but not [^18^F]FDG) was significantly higher in cirrhotic than in non-cirrhotic patients (0.76 ± 0.39 vs. 0.40 ± 0.07, *p* < 0.001). A variable degree of increased [^68^Ga]Ga-FAPI-04 uptake, and with different patterns, was found by Tatar et al. [52] in patients with liver cirrhosis, in the absence of clear [^18^F]FDG distribution abnormalities. No significant differences in 68Ga-FAPI-04 uptake were reported in another study, although a subgroup analysis on cirrhotic patients showed a higher sensitivity in detecting liver lesions using [^68^Ga]Ga-FAPI-04 PET/CT than [^18^F]FDG PET/CT [20]. Lastly, one study concluded that [^68^Ga]Ga-FAPI sensitivity in intrahepatic tumors was not affected by cirrhosis, with a diagnostic accuracy that was higher than that of [^18^F]FDG in cirrhotic patients [95].

#### 2.3.2. Primary Liver Tumors: FAPI PET Diagnostic Performance and Comparison with [^18^F]FDG and/or Other Tracers

Twenty-three studies using [^68^Ga]Ga-FAPI-04, [^68^Ga]Ga-FAPI-46, [^68^Ga]Ga-DOTA-FAPI, [^68^Ga]Ga-DOTA-FAPI-04, Al^18^F-NOTA-FAPI-04, or [^18^F]FAPI as radiopharmaceuticals exclusively focused on primary liver tumors; a 60 min interval from injection to image acquisition ensured high [^68^Ga]Ga-FAPI uptake in lesions, with the median SUVmax of all primary tumors ranging from 2.3 to 15.6 [22,62,65,79,81,82] and the median TBR ranging from 4.8 to 15.9 [22,62,79,81]. These studies mostly included HCC (176 patients) and CC (98 patients, of which ICC was specified for 45 patients), both FAPI-avid malignancies: indeed, the lesions’ SUVmax ranged from 1.89 to 18.89 [62,65] for HCC and from 7.7 to 28.6 [22,62,79] for ICC. Nevertheless, only in patients with HCC was a positive association demonstrated between [^68^Ga]Ga-FAPI uptake and differentiation degree: particularly, poorly differentiated HCCs exhibited a higher uptake than well-differentiated ones [20,22]. At the same time, for cholangiocarcinoma (CC), significantly higher [^68^Ga]Ga-FAPI-46 uptake was observed in Grade 3 than in Grade 2 malignancies [79].

Comparison with another radiopharmaceutical was performed in 19 studies ([^18^F]FDG in all cases); details are reported in Table 3.

Higher radiopharmaceutical uptake was mostly observed in patients with CC than in those with HCC: particularly, in 10 patients with treatment-naïve liver lesions, the [^68^Ga]Ga-FAPI-46-derived SUVmax and TBR were significantly higher in the 5 with CC than in the 5 with HCC (24.02 vs. 7.54 and 21.07 vs. 4.98, respectively; *p* < 0.005) [62]. Similarly, in a prospective assessment of 20 cases with liver lesions suspected of malignancy, [^68^Ga]Ga-FAPI-04-derived SUVmax and TBRmax values were higher in ICC than in HCC (14.14 ± 2.20 vs. 8.47 ± 4.06 for SUVmax; 26.46 ± 4.94 vs. 7.13 ± 5.52 for TBRmax; *p* < 0.05); in the same paper, immunohistochemical analysis of samples from 7 patients with ICC demonstrated very high FAPI expression by stromal cells [20]. In a prospective pilot study including 24 patients with HCC or CC, the median FAPI SUVmax and TBR for primary CC were higher than for HCC as well (CC median SUVmax 17.7 and TBR 7.14, vs. HCCs median SUVmax 11.47 and TBR 2.15) [81]. Furthermore, the combination of higher uptake in lesions and lower physiological background activity in liver parenchyma resulted in a better delineation of liver lesions with [^68^Ga]Ga-FAPI tracers compared to [^18^F]FDG [62,87,91]: [^68^Ga]Ga-FAPI-46 was much more sensitive than [^18^F]FDG in both CC (100% vs. 50%) and HCC patients (100% vs. 71%) [62], and with a higher detection rate (100% vs. 57% in ICC and 94% vs. 69% in HCC) [22].

A significantly higher [^68^Ga]Ga-FAPI uptake than that of [^18^F]FDG was observed in well- and moderately differentiated HCC lesions than in poorly differentiated ones [20,67], and differences in detection performances were more marked for small-sized lesions, which are difficult to detect using [^18^F]FDG due to physiological background activity [67]; nevertheless, some other authors demonstrated a low ability of [^68^Ga]Ga-FAPI PET/CT in detecting lesions < 1.5 cm [22]. In a study by Wang et al. [67] on 25 HCC lesions, although they found a comparable SUVmax and a higher TBR of [^68^Ga]Ga-FAPI-04 compared to [^18^F]FDG, the 15 high-grade (3–4) lesions had significantly higher [^18^F]FDG uptake than the 12 low-grade (1–2) lesions (SUVmax 7.55 ± 3.74 vs. 3.74 ± 0.82, and TBR 3.66 ± 1.57 vs. 1.93 ± 0.44; *p* 0.045), while no significant differences were found for [^68^Ga]Ga-FAPI-04 (SUVmax 6.26 ± 3.67 vs. 4.62 ± 2.10 and TBR 11.90 ± 9.05 vs. 10.17 ± 4.70).

For cholangiocarcinoma, instead, higher [^68^Ga]Ga-FAPI-46 uptake was reported in Grade 3 than Grade 2 malignancies, while no significant difference was observed for [^18^F]FDG [79].

Some papers explored FAPI radiopharmaceuticals in uncommon primary liver malignancies: in a patient with a low-differentiated neuroendocrine liver carcinoma, [^68^Ga]Ga-DOTA-FAPI-04 PET/CT showed a higher uptake than [^18^F]FDG (SUVmax 15.6 vs. 10.5) [66], while [^68^Ga]Ga-FAPI exhibited a higher TBR than [^18^F]FDG (1.6 vs. 0.7) in a primary hepatic extranodal marginal zone MALToma, uncovering the potential of [^68^Ga]Ga-FAPI PET/CT in evaluating liver involvement in lymphoma, especially in non-FDG-avid subtypes [64]. [^68^Ga]Ga-FAPI showed high potential in detecting liver PEComa, thanks to a TBR 20 times higher than that of [^18^F]FDG (23.8 vs. 1.2) [76]. Slightly higher [^68^Ga]Ga-FAPI uptake than that of [^18^F]FDG was found in a patient with hepatic adenocarcinoma (SUVmax 8.9 vs. 6) and concurrent pulmonary false positive nodular alteration due to cryptococcosis [74]. [^68^Ga]Ga-FAPI demonstrated more liver lesions from primary hepatic inflammatory myofibroblastoma than [^18^F]FDG [77].

#### 2.3.3. Liver Metastases: FAPI PET Diagnostic Performance and Comparison with [^18^F]FDG and/or Other Tracers

Thirty-nine studies have analyzed FAPI uptake in liver metastases from various primary neoplasms; particularly, twenty-seven of them compared [^68^Ga]Ga-FAPI and [^18^F]FDG [23,24,25,26,27,29,32,34,35,36,37,38,39,41,42,43,44,45,46,48,49,52,53,57,58,59,60,61], while two compared [^18^F]FAPI and [^18^F]FDG [50,54] and two compared [^68^Ga]Ga-FAPI-04 and [^68^Ga]Ga-DOTATATE [56,57]. The most used FAPI-directed radiopharmaceuticals were [^68^Ga]Ga-FAPI-04 and [^68^Ga]Ga-FAPI-46.

Most selected studies included gastrointestinal tumors as primary lesions, particularly pancreatic [23,26,27,38,55,60], gastric [23,26,37,41,45], colorectal carcinoma [23,26,34,41,42,43,45,54,59], and a case of gastric Kaposi sarcoma [49]; NETs constitute the second largest group of primary neoplasms included [24,31,34,35,39,40,56]. The remaining studies were performed in lung cancer [25,29,34,48], GISTs [44,47], primary extrapulmonary tumors in the chest [53], patients with multiple endocrine neoplasia type 2A syndrome [28], mucosa-associated lymphoid tissue lymphoma [30], uveal malignant melanoma [32], medullary or papillary thyroid carcinomas [33,52], radioiodine-resistant follicular cell-derived thyroid cancer [58], breast cancer [36,61], chromophobe renal cell carcinoma [46], sarcomatoid renal cell carcinoma [51], various lymphomas [57], or unknown primary [50].

In most studies, imaging was obtained 60 min after FAPI tracer injection; in three studies, PET/CT was performed between 30 and 60 min after [^68^Ga]Ga-FAPI administration [37,45,56]. In one study, [^68^Ga]Ga-FAPI-46 uptake was compared between 20 and 60 min post-injection acquisitions in 33 patients with suspected recurrences of pancreatic ductal adenocarcinoma, and similar detection rates were found at both timepoints in seven liver metastases (100%), along with higher SUVmax values than in reactive and cholestatic liver lesions and relatively stable TBR values across timepoints [55]. [^68^Ga]Ga-FAPI uptake was variable among studies, with median SUVmax values ranging from 2.1 to 22.2. Two studies, by Zhang et al. [27] and Qin et al. [37], used PET/MRI for image acquisition. Dynamic PET/CT images lasting 60 min were obtained in two studies: Geist BK et al. [85] performed dynamic images of the livers in 8 patients; Xing et al. [88] performed list-mode scans in 10 patients and sequential whole-body scans at 10 min intervals from 10 to 60 min post-injection in 12 patients (1 of them with liver metastases from sigmoid cancer). Moreover, Hu K et al. [84] performed a whole-body dynamic PET/CT assessment lasting 20 min in four patients with various cancers, for biodistribution assessment.

Comparisons with other radiopharmaceutical(s) were performed in 35 studies: details are reported in Table 4.

In gastrointestinal tumors, [^68^Ga]Ga-FAPI performs better than [^18^F]FDG in detecting liver metastases, overall. Particularly, in four studies, a higher uptake of [^68^Ga]Ga-FAPI than [^18^F]FDG was observed, along with higher [^68^Ga]Ga-FAPI SUVmax values [23,26,37,41]. Furthermore, [^68^Ga]Ga-FAPI detected liver metastases that exhibited no [^18^F]FDG uptake [26,37,38,41,43], including the case of a patient with a pancreatic mass suspected of malignancy and [^68^Ga]Ga-FAPI-positive but [^18^F]FDG-negative liver lesions [38,74]. The sensitivity of [^68^Ga]Ga-FAPI in detecting liver metastases exceeded that of [^18^F]FDG in the study of Sahin E et al. [23] on 31 patients with various gastrointestinal cancers (96.6% for [^68^Ga]Ga-FAPI vs. 70.8% for [^18^F]FDG). In the study by Qin C et al. [37] on three patients with liver metastases from gastric carcinoma, [^68^Ga]Ga-FAPI PET/MRI and [^18^F]FDG PET/CT performed equally (100% patient-based sensitivity), although more lesions were detected by [^68^Ga]Ga-FAPI PET/MRI (13 vs. 6), with a higher normalized SUVmax (5.84 vs. 3.30). However, in a study by Zhang Z et al. [27] on 32 patients with pancreatic adenocarcinoma, [^68^Ga]Ga-FAPI-04 detected a lower number of liver lesions than [^18^F]FDG (104 vs. 181 in 5 patients), with a lower uptake for [^68^Ga]Ga-FAPI-04 (5.97 ± 2.19 vs. 8.64 ± 2.04); in the same study, MRI revealed many more liver micrometastases, but had no influence on the M stage and clinical management. In the study by Lin 2023 [43] on 9 patients with liver metastases from colorectal cancer, despite no significant differences in SUVmax, [^68^Ga]Ga-FAPI had a higher TBR than [^18^F]FDG (3.7 vs. 1.9); Li C et al. [45] found up to 2–3 times higher SUVmax- and SUVmean-derived TBRs and TLRs using [^68^Ga]Ga-FAPI compared with [^18^F]FDG in 10 patients with liver metastases. [^18^F]FAPI-04 performed similarly to [^18^F]FDG in a patient with liver and lung metastases from colorectal cancer [54]. In the study by Dong Y. et al. [59], FAPI PET/CT showed more true positive liver lesions (13 vs. 7), and FAPI uptake was higher than that of [^18^F]FDG (mean SUV max: 9.4 ± 3.0 vs. 4.8 ± 4.4, *p* = 0.014), which was seen as higher mean TBR ratios (13.1 ± 7.5 vs. 3.2 ± 2.5, *p* = 0.001).

In most studies analyzing liver metastases from NETs, a better depiction of liver lesions was observed using [^68^Ga]Ga-FAPI with respect to [^18^F]FDG and/or [^68^Ga]Ga-DOTATATE [35,39,40], due to lower background liver activity. Conversely, in nine patients with liver metastases from NETs of various origin and one patient with pheochromocytoma, Has Simsek et al. demonstrated a lower detection rate using [^68^Ga]Ga-FAPI-04 than with [^68^Ga]Ga-DOTATATE (27/54 vs. 38/54 lesions, respectively), and a significantly lower uptake in all metastatic sites (median SUVmax 5.1 vs. 16.6 for [^68^Ga]Ga-FAPI-04 and [^68^Ga]Ga-DOTATATE, respectively; additionally, flip-flop [^68^Ga]Ga-FAPI-04/[^68^Ga]Ga-DOTATATE uptake was observed in two patients ([^68^Ga]Ga-FAPI-04-positive/[^68^Ga]Ga-DOTATATE-negative primary lesion and vice versa for metastases [56]. Moreover, in a cohort of 13 patients with NETs mostly from pancreatic and intestinal origins and with suspected liver metastases, the FAPI-derived tumor fraction correlated with Ki-67 better than the FDG-derived one (Spearman’s rho 0.770 vs. 0.524), so FAPI is a potentially valuable predictor of Ki-67 status and, ultimately, of lesion aggressiveness [24].

No significant differences in SUVmax between [^68^Ga]Ga-FAPI and [^18^F]FDG were found in patients with liver metastases from lung cancer; however, the [^68^Ga]Ga-FAPI TBR was higher than that of [^18^F]FDG: particularly, a TBR of 11.4 for [^68^Ga]Ga-FAPI was reported, compared to 1.3 for [^18^F]FDG (*p* = 0.027) in 4 liver metastases from lung adenocarcinoma [25]; a higher TBR was found for [^68^Ga]Ga-FAPI (8 vs. 2.9 for [^18^F]FDG) also in 34 patients with lung cancer [29] despite a comparable number of liver lesions (12 with [^18^F]FDG; 11 with [^68^Ga]Ga-FAPI) and SUVmax (6.7 with [^18^F]FDG; 5.9 with [^68^Ga]Ga-FAPI); finally, Can C et al. [48] also demonstrated a higher [^68^Ga]Ga-FAPI-derived TBR compared to that of [^18^F]FDG (5.40 vs. 2.36; *p* < 0.001) in 19 and 18 liver metastases, irrespective of the primary tumor histology.

Eleven studies (two prospective, one retrospective, and eight case reports) [28,30,32,33,36,44,46,47,50,52,53] have compared the diagnostic performances of FAP-specific radiopharmaceuticals against other tracers in patients with various neoplasms. Particularly, [^68^Ga]Ga-FAPI detected more liver lesions than [^18^F]FDG (77 vs. 30), with significantly higher TBRs (median 9.2 vs. 2; *p* < 0.001), despite comparable SUVmax values (median 9.2 vs. 6.1; *p* = 0.409), in 20 prospectively included women with breast cancer [36]. [^68^Ga]Ga-DOTA.SA.FAPi uptake (TBR) in 17 patients with liver metastases from breast cancer was significantly higher (*p* < 0.05) than that of [^18^F]FDG [61]. Comparable results were obtained by Wu C et al. [47] using [^18^F]FAPI-42 in 35 patients with recurrent or metastatic GIST: a higher number of liver metastases were detected compared to [^18^F]FDG (42 vs. 16), with significantly higher TBRs (median 2.4 vs. 0.9; *p* < 0.001). Moreover, Al^18^F-NOTA-FAPI showed a hepatic lesion of MALT lymphoma not visible on [^18^F]FDG [30], while [^68^Ga]Ga-FAPI-04 depicted liver lesions more clearly (higher number and higher TBR) than [^18^F]FDG in two patients with chromophobe renal cell carcinoma [46] and gastric GIST [44]. More liver metastases from primary extrapulmonary chest tumors, and with a higher tumor-to-liver ratio, were found using [^68^Ga]Ga-FAPI-04 than with [^18^F]FDG (SUVmax was comparable for both tracers) [53]. Conversely, a higher uptake using [^18^F]FDG was found in multiple liver metastases (SUVmax 8.3 vs. 6.5 of [^68^Ga]Ga-FAPI-4) from uveal malignant melanoma in one patient; the same result was found for liver metastases from various lymphomas, with the [^68^Ga]Ga-FAPI SUVmax lower than the [^18^F]FDG SUVmax (respectively, 5.9 vs. 11.6) [57], and the [^68^Ga]Ga-FAPI-4 uptake in liver metastases was even lower than that observed in bilateral knee osteoarthritis [32]. However, liver metastases from medullary thyroid carcinoma were clearly recognizable only using [^68^Ga]Ga-FAPI-04 (only inhomogeneous uptake in liver parenchyma with [^68^Ga]Ga-DOTATATE) [33]. The use of [^68^Ga]Ga-DOTA.SA.FAPi resulted in a higher detection rate for liver metastases (100% vs. 81.3%, *p* < 0.0001) from radioiodine-resistant follicular cell-derived thyroid cancer compared to [^18^F]FDG [58]. Lastly, in a patient with MEN-2A, Barashki S et al. [28] reported a more intense uptake using [^68^Ga]Ga-FAPI-46 than [^18^F]FDG in multiple metastatic liver lesions (although with a similar distribution), and, based on [^68^Ga]Ga-FAPI-46 PET/CT positivity, the patient received [^177^Lu]Lu-FAPI-46 tumor-targeted treatment, with a complete resolution of complaints.

Finally, four studies [31,42,51,55] used FAP-specific radiopharmaceuticals for liver metastases without comparing them against any other tracers. Specifically, Dendl K et al. [31] demonstrated a high [^68^Ga]Ga-FAPI uptake in liver metastases (SUVmax 9.8), comparable to that of peritoneal metastases, and an excellent TBR (8.74) with respect to the surrounding healthy parenchyma. Koerber SA et al. [42] found a high [^68^Ga]Ga-FAPI uptake in 14 liver metastases from low gastrointestinal tract tumors (SUVmax 9.54 ± 3.74; SUVmean 4.86 ± 2.14); in these patients, the [^68^Ga]Ga-FAPI-derived SUVmax and SUVmean in liver metastases were higher than in other distant sites, having been exceeded only by the primary tumor. Hoppner et al. [55] found a high TBR for liver lesions at both 20 and 60 min post-injection acquisition timepoints, although lesions were slightly more discernible in the 60 min scan due to reduced background activity.

When dynamic PET was performed, a variable uptake over time in healthy organs and liver lesions was observed. Particularly, Xing et al. [88] showed a [^68^Ga]Ga-FAPI uptake decrease in healthy organs in the first 30 min, which occurred more slowly 30–60 min post-injection: in both protocols, the liver lesion SUVmax reached its highest value 10 min after injection (8.78 ± 3.30 in protocol 1; 12.69 ± 13.05 in protocol 2), then it was stable until 30 min post-injection (7.60 ± 4.00 in protocol 1; 11.05 ± 4.59 in protocol 2); finally, it decreased slowly from 30 to 60 min (only in protocol 2); similarly, the TBR increased from 10 to 30 min and was relatively stable in the subsequent 30 min. Furthermore, PET frames acquired at 30 and 60 min post-injection allowed researchers to detect all 56 liver lesions (12 were missed at the 10 min post-injection scan), therefore suggesting that it is appropriate to perform scans 30–60 min post-injection. However, as demonstrated by Geist et al. [85], a dynamic PET scan lasting 60 min of the upper abdomen was not able to distinguish among various malignant entities (although they showed a higher uptake than the surrounding tissue), nor was it able to distinguish inflammatory disease from normal liver parenchyma, but only to discriminate between malignant and inflammatory lesions.

#### 2.3.4. FAPI PET False Positive and False Negative Findings

Concerning the interpretation of FAPI PET/CT images, attention should be paid to false positive or negative findings. Although FAP is expressed in malignant lesions’ CAFs, some benign conditions exhibit an increased uptake as well [54,65,74,75,96,97,98,99], leading to possible false positives. Non-neoplastic conditions such as inflammation after wound healing may activate the biologically quiescent fibroblasts to support immune cells and tissue cell proliferation, as part of the damage repair mechanism [100,101,102]; such activation is self-limiting once the damaging event has ceased. However, tumors may induce a persistent fibroblast activation by mimicking inflammatory cells, as “wounds that do not heal” [102,103]: in this process, CAFs contribute to tumor growth by producing pro-inflammatory cytokines and chemokines [3] that support tumor cells’ proliferation, migration, invasion, angiogenesis, and resistance to therapies [3,104].

In a study by Zheng et al. [65] including 182 patients with various malignancies (among them 4 HCC and 2 ICC), inflammatory/infectious conditions, such as osteoarthritis, enthesopathy, periodontitis, mastoiditis, chronic pancreatitis, esophagitis, appendicitis, prostatitis, inflammatory lymph nodes, pneumonia, post-operative changes, and tuberculosis, increased the [^68^Ga]Ga-FAPI-04 uptake. A case of pulmonary cryptococcosis with intense [^68^Ga]Ga-FAPI and [^18^F]FDG uptake, and with a concurrent hepatic adenocarcinoma, was recently reported [74]; moreover, Klebsiella Pneumoniae invasion syndrome mimicking metastatic [^68^Ga]Ga-FAPI-positive dissemination was recorded [99]. Some authors also reported false positive findings related to treatment-induced inflammation in liver parenchyma: particularly, in a study by Siripongsatian et al. [62], two false positives were found with [^68^Ga]Ga-FAPI-46 after trans-arterial chemoembolization in patients with HCC (MRI confirmed post-treatment fibrosis); Chen et al. [71] reported a wedge-shaped area of [^18^F]FAPI uptake around a non-FDG-avid HCC lesion, determined by stromal fibrosis after radiofrequency ablation; more recently, Zhang et al. [97] reported intense [^18^F]FAPI uptake in a case of radiation-induced liver injury after treatment of invasive ductal carcinoma of the breast (the lesion was negative with [^18^F]FDG). Increased [^68^Ga]Ga-FAPI-04 uptake could be related to post-surgical inflammation; therefore, high attention should be paid to not misdiagnose these cases as tumor relapse [22,67]. Other benign diseases, such as lung granuloma, pulmonary infection, thyroid adenoma, angiomyolipoma, arteriovenous malformations, biliary obstruction, liver abscesses, focal nodular hyperplasia, or inflammatory liver nodules, may produce incidental false positive findings [22,55,59,60,65,98].

False negative findings are more frequently observed in patients with liver cirrhosis, which causes diffuse increased FAPI background liver uptake. In a patient with poorly differentiated HCC and liver cirrhosis, the uptake of [^68^Ga]Ga-DOTA-FAPI-04 was only mild in the lesion (SUVmax 2.3) and very intense in the surrounding liver parenchyma (SUVmax 11.1) [70]. Moreover, two false negative (but [^18^F]FDG-positive) bone metastases are described in a patient with HCC [62], possibly due to FAP expression heterogeneity.

## 3. Discussion

The in-human use of radiolabeled FAPI compounds in primary and metastatic liver lesions has been rapidly growing over time, especially thanks to the usually physiological low background uptake in liver parenchyma, which makes lesion detection easier.

The available literature demonstrated the excellent diagnostic performance of FAPI PET/CT in the most common primary liver malignancies, like HCC and ICC: particularly, the high lesion uptake coupled with lower liver background uptake resulted in a higher sensitivity when compared with more “conventional” radiopharmaceuticals like [^18^F]FDG. The TBR, which is affected by low background uptake, is the most strikingly helpful among PET semi-quantitative parameters in recognizing liver lesions [22,62]. Additional intrahepatic lesions may be detected using FAPI PET/CT, leading to a more accurate staging, a better definition of prognosis, and guidance of therapeutic management [22]. FAPI uptake is determined by CAFs expression in the tumor stroma: this explains the overall higher uptake in ICC than in HCC (as confirmed by CAFs’ prevalence over malignant cells in the histological specimens) [22,62]. Moreover, when considering histological degree in HCC patients, a lower uptake was observed in well-differentiated malignancies than in poorly differentiated ones [20], and in hematoxylin and eosin stains, it was demonstrated that more mesenchymal cells are present in less-differentiated lesions [22]. A higher proportion of tumor-associated fibroblasts is a possible explanation for the intense [^68^Ga]Ga-FAPI uptake noticed in fibrolamellar HCC [72]. Recent studies have reported a significantly lower physiological liver background for [^68^Ga]Ga-FAPI than for [^18^F]FDG PET/CT [5,34,78,93]: particularly, Giesel et al. [34] found a normal liver mean SUVmax of 1.42 using [^68^Ga]Ga-FAPI and of 3.10 using [^18^F]FDG with 71 oncological patients. However, an increase in FAPI uptake may be recognized in the setting of liver cirrhosis, in terms of SUV and TBR [22,67]: this is explained by the stimulation of fibroblast activation, leading to the formation of fibrotic scars [105,106]. Hepatitis may also contribute to a higher FAPI liver background uptake, especially in benign regenerative nodules; this could impact the correct detection of malignant lesions [22]. The detection of cirrhosis is important in liver cancer management since it limits the spectrum of possible therapeutic strategies [22,52]; nonetheless, small-sized lesions may be missed where there is a high FAPI background uptake in cirrhotic patients [39,65,67]; we should also consider that small-sized lesions may show an absolutely lower uptake than larger-sized ones (lower hypoxia, which translates into lower FAP expression in CAFs) [67]. Furthermore, FAPI PET/CT exhibited a higher detection rate than [^18^F]FDG PET/CT, due to higher uptake in lesions and a higher TBR, making it possible to use FAPI as a promising agent for hepatic tumor staging [22]. The higher expression of CAFs in the tumor stroma at early disease stages in well- and moderately differentiated lesions involves a high sensitivity of FAPI, in a clinical setting usually characterized by low [^18^F]FDG intra-cell trapping due to enhanced glucose-6-phosphatase activity [39,67]. Moreover, other less common primary liver malignancies exhibiting low [^18^F]FDG uptake, such as neuroendocrine liver carcinoma, marginal MALT lymphoma, or PEComa, or even less common borderline primary liver lesions such as inflammatory myofibroblastoma, may benefit from the use of FAPI PET/CT, especially in patients exhibiting doubtful lesions with conventional imaging [64,66,73,76,77].

^68^Ga-labeled FAPI radiopharmaceuticals have demonstrated excellent performances in identifying liver metastases from various gastrointestinal cancers: particularly, in the vast majority of the available literature, [^68^Ga]Ga-FAPI has proven to find more liver lesions, and with higher SUVmax values, compared with [^18^F]FDG [23,26,37,38,41,43]; moreover, [^68^Ga]Ga-FAPI’s sensitivity to liver metastases is higher than that of [^18^F]FDG in this setting [23]. However, in patients with pancreatic cancer, [^68^Ga]Ga-FAPI PET/MR detected a lower number of liver metastases than [^18^F]FDG, and with a lower uptake intensity [27]. The TBR, as the ratio between the maximum or mean uptake of the lesion and that of background (usually physiological liver) activity, may be 2–3 times higher for [^68^Ga]Ga-FAPI PET/CT than for [^18^F]FDG in liver metastases from colorectal cancer (although some overlap was found) [43,45]. Even in liver metastases from lung cancer, [^68^Ga]Ga-FAPI PET/CT performs better than [^18^F]FDG in terms of TBR (higher), despite similar SUVmax values [25,29,48]. Concerning liver metastases from NETs, [^68^Ga]Ga-FAPI detects more lesions than [^18^F]FDG [35,39] because of the lower liver background activity; more controversial is the comparison between [^68^Ga]Ga-FAPI and [^68^Ga]Ga DOTATATE [35,40,56], because of a reported lower detection rate in liver metastases from NETs when using [^68^Ga]Ga-FAPI [56]. Moreover, in patients with pancreatic and intestinal NETs, FAPI PET/CT has been demonstrated to be a potentially valuable predictor of lesion aggressiveness and risk of progression, considering the very good correlation between FAPI PET/CT-derived tumor fractions and Ki-67, which is much better than the [^18^F]FDG-derived one [24]. FAPI radiopharmaceuticals perform better than [^18^F]FDG ones, showing significantly higher TBRs and, in some cases, also more liver lesions otherwise not evident with [^18^F]FDG, in breast cancer [36], GISTs [44,47], a case of MALT lymphoma [30], chromophobe renal cell carcinoma [46], and a case of MEN-2A syndrome [28]. In a woman with medullary thyroid carcinoma, Kuyumcu S et al. [33] reported focal liver metastases only using [^68^Ga]Ga-FAPI-04, whereas [^68^Ga]Ga-DOTATATE showed only subtle inhomogeneity in the liver parenchyma uptake. Conversely, in some cases, such as in a reported uveal malignant melanoma [32], FAPI radiopharmaceutical uptake in liver metastases may be lower than that of [^18^F]FDG (even lower than in an incidental inflammatory condition such as knee osteoarthritis). Two papers [14,42] have explored the diagnostic performance of ^68^Ga-labeled FAP-directed radiopharmaceuticals in detecting liver metastases, not compared to other tracers: in these studies, a high [^68^Ga]Ga-FAPI uptake was evident in liver metastases from head-and-neck, gastrointestinal, biliary–pancreatic, urinary tract, and neuroendocrine tumors, with high TBRs with respect to the surrounding healthy parenchyma [14,42], and liver metastases derived from low gastrointestinal tract tumors exhibited the highest [^68^Ga]Ga-FAPI-04 or [^68^Ga]Ga-FAPI-46 uptake among metastatic sites (exceeded only by the primary tumors) [42].

Dynamic acquisition has been demonstrated to be useful in depicting liver metastases, although proper acquisition timing has to be addressed for better delineation of lesions with respect to the surrounding parenchyma: the available studies [85,88] found that most lesions are most visible 30–60 min after radiopharmaceutical administration, a timing that ensures the highest TBR and an adequate lesion SUVmax (on the other hand, only a few lesions are visible in the first 10 min). However, despite FAPI PET/CT SUVmax values attained during dynamic acquisition being overall higher in malignant lesions than in normal liver parenchyma or in benign lesions, FAPI kinetics are not able to distinguish among different histologies of primary liver lesions (HCC vs. non-HCC) or between liver metastases and primary lesions, nor benign lesions from normal parenchyma.

There is recent evidence of the usefulness and safety of FAPI-directed beta-emitting compounds such as [^177^Lu]Lu-FAPI in the treatment of liver metastases from various histologies, using FAPI PET/CT for suitability assessment and post-therapy evaluation. Barashki et al. [28] described a patient with MEN type 2A and multi-site metastases (including liver, bone, and lymph nodes) who underwent [^177^Lu]Lu-FAPI-46 therapy on [^68^Ga]Ga-FAPI-46 positive lesions on PET/CT, resulting in clinical improvement (resolution of abdominal pain). Baum et al. [12] have performed the first in-human study on peptide-targeted radionuclide therapy using ^177^Lu-FAP-2286 to treat ^68^Ga-FAP-2286- or [^68^Ga]Ga-FAPI-04-positive liver metastases from breast, pancreas, rectum, or ovary adenocarcinomas, reporting high ^177^Lu-FAP-2286 uptake in tumor lesions up to 10 days after treatment (3.0 ± 2.7 Gy/GBq), yet no Grade 4 adverse events (most patients complained of self-limiting headaches, temporary flare-up of the abdominal pain, anemia, or pancytopenia).

The first limitation to be considered for this review is the small cohort size in most included studies: indeed, although patient recruitment designs were mostly prospective, 47 out of the 76 papers included fewer than 15 patients, while 19 studies included a larger population with liver malignancies [20,21,22,23,58,61,62,67,78,80,81,82,86,88,89,90,93,94,95]. Therefore, future trials/studies on a larger population are needed. Heterogeneity in clinical and technical aspects was also observed: most studies included a mixed population, with primary liver lesions of different histologies (HCC, ICC, or less common ones), or with liver metastases from different primary cancers (mostly gastrointestinal); moreover, different FAPI-directed radiopharmaceuticals (either 68Ga- or 18F-labeled), among studies or in the same paper, possibly with slight differences in pharmacokinetics or pharmacodynamics (affecting biodistribution in healthy organs and malignant lesions, preferred acquisition timing for best detection rate, and effective dose), were employed. Particularly, [^68^Ga]Ga-FAPI-02 is selectively accumulated in FAP-expressing tissue, with a significantly higher uptake than that of [^18^F]FDG in malignant lesions; [^68^Ga]Ga-FAPI-04 is rapidly internalized into FAP-positive tumors, showing fast clearance and a very rapid accumulation in tumor sites (10 min after administration) (tumor uptake is even higher and more rapid using [^68^Ga]Ga-FAPI-46 compared to [^68^Ga]Ga-FAPI-04); [^68^Ga]Ga-DOTA.SA.FAPI shows an advantage in brain metastasis compared with [^18^F]FDG; [^18^F]F-NOTA-FAPI-04 exhibits no significant defluorination during image acquisition; therefore, it shows a higher uptake and TBR in liver and bone tumors compared with [^68^Ga]Ga-FAPI. Moreover, dynamic PET scans were not performed using unique protocols, and in some studies, PET/MRI was used in addition or in place of PET/CT.

## 4. Materials and Methods

### 4.1. Literature Search and Information Sources

A comprehensive computer literature search of the PubMed and MEDLINE databases was conducted to find relevant published articles on the diagnostic use of PET with radiolabeled FAPIs in patients with neoplastic liver lesions, including both primary liver neoplasms (such as hepatocellular carcinoma and cholangiocarcinoma) and liver metastases from any other primary malignancy. A search algorithm based on the combination of the terms (“FAPI”) AND (“hepatic” OR “liver”) was used. The last update of the literature search was 1st January 2024. Titles and abstracts of the retrieved studies were screened independently by two researchers (FM and GP).

### 4.2. Inclusion and Exclusion Criteria and Data Analysis

Only articles written in the English language and studies conducted on human subjects, that included patients with neoplastic liver lesions (primary hepatic tumors and/or liver metastases), and with available data on the diagnostic performance of PET imaging with radiolabeled FAPIs (regardless of disease stage and clinical indication for PET examination) were considered. After screening titles and abstracts, an initial selection was performed, excluding (a) articles not within the field of interest of this review (e.g., not evaluating patients with neoplastic liver lesions; with no data on PET imaging; focused on FAPI synthesis, biodistribution, pharmacokinetics, or dosimetry, or only on radiolabeled FAPI therapy); (b) pre-clinical studies; (c) review articles and meta-analyses, editorials, letters, commentaries, and conference proceedings. Then, two researchers (FM and GP) independently reviewed the full-text versions of the selected articles to assess them for final eligibility, and possible disagreements were resolved by consensus. For each included article, information was collected about the basic clinical characteristics (study design, number of patients with neoplastic liver lesions, primary liver tumors, or liver metastases, etc.), and methodological aspects (FAPI tracer, injected activity, scan delay as the interval between injection and PET image acquisition, comparison with PET imaging using other radiopharmaceuticals when available, etc.).

## 5. Conclusions

From the present review, it has emerged that the use of radiolabeled FAPI PET/CT for the study of primary and/or metastatic liver lesions, despite the evident heterogeneity in population size and in study design of the included articles, as well as in the radiopharmaceuticals employed and in histology among primary and metastatic liver lesions, carries excellent diagnostic performance, usually better than “conventional” radiopharmaceuticals: the significantly lower FAPI liver background uptake resulted in a higher sensitivity of this imaging modality in both primary and metastatic liver lesions, especially when compared to [^18^F]FDG.

## Figures and Tables

**Figure 1 ijms-25-07197-f001:**
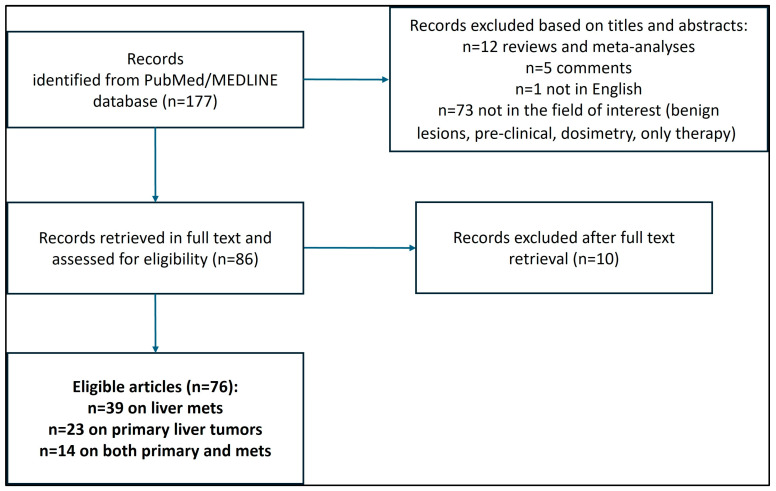
Algorithm for article retrieval and selection.

**Table 1 ijms-25-07197-t001:** Basic, clinical, and methodological characteristics of 23 articles on primary liver tumors.

First Author	Year	Journal	Country	Study Design	Histology	Clinical Indication for FAPI Imaging	Patients with Malignant Liver Lesions (n)	FAPI PET Modality	FAPI Tracer	FAPI Activity	Post-inj. Delay (min)	FOV	Other Tracer(s)
Li T [77]	2023	Clin Nucl Med	China	P (trial)	inflammatory myofibroblastoma	staging	1	PET/CT	[^68^Ga]Ga-FAPI (nos.)	n.r.	n.r.	vertex to upper thigh	[^18^F]FDG
Zhang J [78]	2023	Cancer Imaging	China	P	HCC	staging, restaging	67	PET/CT	[^18^F]FAPI (nos.)	n.r.	n.r.	vertex to upper thigh	[^18^F]FDG
Li Z [76]	2023	Clin Nucl Med	China	P (trial)	PEComa	lesion characterization	1	PET/CT	[^68^Ga]Ga-FAPI (nos.)	n.r.	n.r.	vertex to upper thigh	[^18^F]FDG
Al-Ibraheem A [75]	2023	Clin Nucl Med	Jordan	n.r.	ICC	staging	1	PET/CT	[^68^Ga]Ga-FAPI (nos.)	n.r.	n.r.	vertex to toes	NO
Liu M [74]	2023	Clin Nucl Med	China	P (trial)	hepatic adenocarcinoma	lesion characterization	1	PET/CT	[^68^Ga]Ga-FAPI (nos.)	n.r.	n.r.	vertex to upper thigh	[^18^F]FDG
Suthar RR [72]	2023	Clin Nucl Med	India	n.r.	fibrolamellar HCC	lesion characterization	1	PET/CT	[^68^Ga]Ga-FAPI (nos.)	n.r.	n.r.	vertex to upper thigh	[^18^F]FDG
Ou L [73]	2023	Clin Nucl Med	China	n.r.	inflammatory myofibroblastoma	lesion characterization	1	PET/CT	[^68^Ga]Ga-FAPI (nos.)	n.r.	n.r.	n.r.	NO
Wu M [82]	2023	J Nucl Med	China	P	HCC	survival prediction	22	PET/CT	[^68^Ga]Ga-FAPI-04	2.22–2.96 MBq/kg	42–90	vertex to middle thigh	[^18^F]FDG
Rajaraman V [81]	2023	Clin Nucl Med	India	P	HCC CC	staging	24	PET/CT	[^68^Ga]Ga-FAPI-04	185–370 MBq	60	vertex to upper thigh	[^18^F]FDG
Jinghua L [80]	2023	Eur J Nucl Med Mol Imaging	China	P (trial)	CC	lesion characterization, staging, restaging	38 (including 22 ICC))	PET/CT	[^68^Ga]Ga-DOTA-FAPI	2.04 ± 0.22 MBq/kg	40–60	vertex to upper thigh	[^18^F]FDG
Pabst KM [79]	2023	J Nucl Med	Germany	P (trial)	CC	staging, restaging	10 (including 6 ICC)	PET/CT	[^68^Ga]Ga-FAPI-46	89 MBq (IQR 79–128)	15 (IQR 10–38)	vertex to upper thigh	[^18^F]FDG
Yao X [68]	2023	Clin Nucl Med	China	P (trial)	n.r.	staging	1	PET/MRI	[^68^Ga]Ga-FAPI (nos.)	n.r.	n.r.	vertex to upper thigh	[^18^F]FDG
Zhou Y [70]	2023	Eur J Nucl Med Mol Imaging	China	P (trial)	HCC	staging	1	PET/CT	[^68^Ga]Ga-FAPI-04	n.r.	n.r.	vertex to upper thigh	NO
Pan B [69]	2022	Hell J Nucl Med	China	P (trial)	HCC	lesion characterization	1	PET/CT	Al^18^F-NOTA-FAPI-04	n.r.	n.r.	vertex to upper thigh	[^18^F]FDG
Chen D [71]	2022	Cancer Manag Res	China	n.r.	HCC	restaging	1	PET/CT	[^18^F]-FAPI (nos.)	n.r.	n.r.	skull base to mid-thigh	[^18^F]FDG
Siripongsatian D [62]	2022	Mol Imaging Biol	Thailand	R	HCC ICC	staging, restaging	27	PET/CT + liver PET/MRI	[^68^Ga]Ga-FAPI-46	2.59 MBq/kg	60	vertex to mid-thigh	[^18^F]FDG
Pang Y [64]	2022	Clin Nucl Med	China	P (trial)	primary hepatic extranodal marginal zone lymphoma of MALT	staging	1	PET/CT	[^68^Ga]Ga-FAPI (nos.)	n.r.	n.r.	vertex to upper thigh	[^18^F]FDG
Ergül N [66]	2022	Clin Nucl Med	Turkey	P (trial)	low-differentiated NEC	lesion characterization	1	PET/CT	[^68^Ga]Ga-DOTA-FAPI-04	n.r.	n.r.	vertex to upper thigh	[^18^F]FDG
Siripongsatian D [63]	2021	Nucl Med Mol Imaging	Thailand	n.r.	ICC	restaging	1	PET/CT + upper abdomen PET/MRI	[^68^Ga]Ga-FAPI-46	n.r.	60	n.r.	[^18^F]FDG
Zheng S [65]	2021	Ann Nucl Med	China	R	HCC ICC	lesion characterization	6	PET/CT	[^68^Ga]Ga-FAPI-04	3.7 MBq/kg	30–60	vertex to upper thigh	NO
Wang H [67]	2021	Front Oncol	China	R	HCC	staging, restaging	25	PET/CT	[^68^Ga]Ga-FAPI-04	~185 MBq	60	skull base to upper thigh	[^18^F]FDG
Guo W [22]	2020	Eur J Nucl Med Mol Imaging	China	R	HCC ICC	lesion characterization, staging, restaging	32	PET/CT	[^68^Ga]Ga-FAPI-04	148–259 MBq	60	vertex to upper thigh	[^18^F]FDG
Shi X [20]	2020	Eur J Nucl Med Mol Imaging	China	P	HCC ICC	lesion characterization	17	PET/CT	[^68^Ga]Ga-FAPI-04	3.59 MBq/kg	40–50	vertex to upper thigh	[^18^F]FDG

Abbreviations: P—prospective, R—retrospective, FOV—field of view, NEC—neuroendocrine carcinoma, CC—cholangiocarcinoma, HCC—hepatocellular carcinoma, ICC—intrahepatic cholangiocarcinoma, MALT—mucosa-associated lymphoid tissue, n.r.—not reported, nos.—not otherwise specified.

**Table 2 ijms-25-07197-t002:** Basic, clinical, and methodological characteristics of 39 articles on liver metastases.

First Author	Year	Journal	Country	Study Design	Site of Primary Tumor	Clinical Indication for FAPI Imaging	Patients with Malignant Liver Lesions (n)	FAPI PET Modality	FAPI Tracer	FAPI Activity	Post-inj. Delay (min)	FOV	Other Tracer(s)
Koerber SA [42]	2020	J Nucl Med	Germany	R	Lower gastrointestinal	lesion characterization, staging	14	PET/CT	[^68^Ga]Ga-FAPI-04; [^68^Ga]Ga-FAPI-46	111–298 MBq	60	vertex to upper thigh	[^18^F]FDG
Cheng Z [35]	2021	Clin Nucl Med	China	P (trial)	Pancreatic NET	lesion characterization and search for primary tumor	1	PET/CT	[^68^Ga]Ga-FAPI (nos.)	n.r.	n.r.	vertex to upper thigh	[^18^F]FDG, [^68^Ga]Ga-DOTATATE
Deng M [38]	2021	Clin Nucl Med	China	P (trial)	Pancreatic carcinoma	staging	1	PET/CT	[^68^Ga]Ga-FAPI (nos.)	n.r.	n.r.	vertex to upper thigh	[^18^F]FDG
Dendl K [31]	2021	Eur J Nucl Med Mol Imaging	Germany	R	Mixed	staging, therapy evaluation, biodistribution	n.r.	PET/CT	[^68^Ga]Ga-FAPI-04 (n = 22); [^68^Ga]Ga-FAPI-46 (n = 20); [^68^Ga]Ga-FAPI-74 (n = 13)	252 MBq (range 118–340)	60	vertex to upper thigh	NO
Giesel FL [34]	2021	Eur J Nucl Med Mol Imaging	Germany, USA, South Africa	R	Mixed	staging, biodistribution	n.r.	PET/CT	[^68^Ga]Ga-FAPI-02 (n = 6); [^68^Ga]Ga-FAPI-04 (n = 32); [^68^Ga]Ga-FAPI-46 (n = 32); [^68^Ga]Ga-FAPI-74 (n = 1)	median 185 MBq (range 52–325)	60	vertex to upper thigh	[^18^F]FDG
Kömek H [36]	2021	Ann Nucl Med	Japan	P	Breast cancer	staging, restaging	3	PET/CT	[^68^Ga]Ga-FAPI-04	2 MBq/kg	60	vertex to upper thigh	[^18^F]FDG
Kömek H [40]	2021	Clin Nucl Med	Turkey	P (trial)	Pancreatic NET	staging, evaluation before targeted therapy	1	PET/CT	[^68^Ga]Ga-FAPI (nos.)	n.r.	n.r.	vertex to upper thigh	[^68^Ga]Ga-DOTATATE
Kreppel B [24]	2021	Nuklearmedizin	Germany	R	NET (pancreas = 8; small bowel = 2; appendix = 1; primary site unknown = 2)	lesion characterization, prognosis stratification	13	PET/CT	[^68^Ga]Ga- DATA5m.SA.FAPi	184 ± 22 MBq	79 ± 34	vertex to upper thigh	[^18^F]FDG and [^68^Ga]Ga-DOTA-TOC
Kuyumcu S [33]	2021	Endocrine	Turkey	n.r.	Medullary thyroid carcinoma	restaging	1	PET/CT	[^68^Ga]Ga-FAPI-05	n.r.	n.r.	vertex to upper thigh	[^68^Ga]GaDOTATATE
Pang Y [41]	2021	Radiology	China	R	Gastric and colorectal cancers	staging, restaging	10	PET/CT	[^68^Ga]Ga-FAPI (nos.)	1.8–2.2 MBq/kg	60	vertex to upper thigh	[^18^F]FDG
Sahin E [23]	2021	Eur J Radiol	Turkey	R	Colorectal = 15; pancreas = 9; gastric = 4; other = 3	staging, restaging	29	PET/CT	[^68^Ga]Ga-FAPI-04	2–3 MBq/kg	45	vertex to upper thigh	[^18^F]FDG
Wang H [39]	2021	Eur J Nucl Med Mol Imaging	China	R	Pancreatic NET	lesion characterization	1	PET/CT	[^68^Ga]Ga-FAPI-04	n.r.	n.r.	skull base to upper thigh	[^18^F]FDG and [^11^C]acetate
Barashki S [28]	2022	Clin Nucl Med	Iran	P (trial)	MEN 2A	staging and targeted therapy eligibility	1	PET/CT	[^68^Ga]Ga-FAPI-46	n.r.	n.r.	vertex to upper thigh	[^68^Ga]Ga-DOTATATE and [^131^I]mIBG
Can C [48]	2022	Nucl Med Commun	Turkey	R	Lung cancer (NSCLC)	staging	7	PET/CT	[^68^Ga]Ga-FAPI-04	2 Mbq/kg	55–65	vertex to mid-thigh	[^18^F]FDG
Elboga U [26]	2022	Mol Imaging Biol	Turkey	R	CCR, gastric cancer, pancreaticobiliary cancer	staging (with known peritoneal involvement)	n.r.	PET/CT	[^68^Ga]Ga-FAPI-04	2 MBq/kg	60	vertex to mid-thigh	[^18^F]FDG
Erol Fenercioğlu OE [32]	2022	Clin Nucl Med	Turkey	P (trial)	Uveal melanoma	restaging	1	PET/CT	[^68^Ga]Ga-FAPI-4	278 MBq	n.r.	vertex to upper thigh	[^18^F]FDG
Kou Y [30]	2022	Clin Nucl Med	China	P (trial)	MALT lymphoma of sub-mandibular gland	staging	1	PET/CT	Al^18^F-NOTA-FAPI-04	n.r.	n.r.	vertex to upper thigh	[^18^F]FDG
Qin C [37]	2022	J Nucl Med	China	P	Gastric cancer	staging, restaging	3	PET/MRI	[^68^Ga]Ga-FAPI-04	1.85–3.7 MBq/kg	30–60	vertex to upper thigh	[^18^F]FDG
Tatar G [49]	2022	Clin Nucl Med	Turkey	P (trial)	Gastric Kaposi sarcoma	staging	1	PET/CT	[^68^Ga]Ga-FAPI-4	n.r.	60	vertex to mid-thigh	[^18^F]FDG
Wang L [29]	2022	Radiology	China	P	Lung cancer	staging, restaging	n.r.	PET/CT	[^68^Ga]Ga-FAPI-47	n.r.	n.r.	vertex to upper thigh	[^18^F]FDG
Wu C [47]	2022	Eur J Nucl Med Mol Imaging	China	R	GIST	restaging	n.r.	PET/CT	[^18^F]FAPI-42	259 ± 26 MBq	60	vertex to mid-thigh	[^18^F]FDG
Wu J [25]	2022	Front Oncol	China	P	Lung cancer (NSCLC)	staging	2	PET/CT	[^68^Ga]Ga-FAPI (nos.)	1.85–2.59 MBq/kg	60 + M2:M22	vertex to the upper portion of the mid-thigh	[^18^F]FDG
Xie F [46]	2022	Clin Nucl Med	China	n.r.	Renal cell carcinoma	restaging	1	PET/CT	[^68^Ga]Ga-FAPI-04	n.r.	n.r.	vertex to upper thigh	[^18^F]FDG
Zhang Z [27]	2022	Eur J Nucl Med Mol Imaging	China	P	Pancreatic carcinoma	lesion characterization, staging	5	PET/MRI	[^68^Ga]Ga-FAPI-04	1.85–3.70 MBq/kg	n.r.	vertex to mid-thigh	[^18^F]FDG
Li C [45]	2023	Eur Radiol	China	R (post hoc of larger P study)	Gastrointestinal carcinoma (gastric = 28; CCR = 21; appendix = 2)	staging, restaging	10	PET/CT	[^68^Ga]Ga-FAPI-04	1.85–3.70 MBq/kg	30–60	vertex to upper thigh	[^18^F]FDG
Lin X [43]	2023	Front Oncol	China	P	Colorectal cancer	staging, restaging, post-treatment	9	PET/CT	[^68^Ga]Ga-FAPI-04	1.85–2.96 MBq/kg	60	vertex to upper thigh	[^18^F]FDG
Zhang Z [44]	2023	Clin Nucl Med	China	P	GIST	staging	1	PET/CT	[^68^Ga]Ga-FAPI-04	n.r.	n.r.	vertex to upper thigh	[^18^F]FDG
Ballal S [61]	2023	Pharmaceuticals	India	R	Breast cancer	staging, restaging	17	PET/CT	[^68^Ga]Ga-DOTA.SA.FAPi	200 MBq	60	vertex to upper thigh	[^18^F]FDG
Liu Q [60]	2023	Eur Radiol	China	R	Pancreatic cancer	lesion characterization, staging, restaging	n.r.	PET/CT	[^68^Ga]Ga-DOTA-FAPI-04	1.8–2.2 MBq/kg	60	skull base to upper thigh	[^18^F]FDG
Dong Y [59]	2023	Nucl Med Commun	China	R (post hoc of larger P study)	Colorectal cancer	staging	5	PET/CT	[^68^Ga]Ga-FAPI-04, [^18^F]FAPI-42	0.04–0.06 mCi/kg	60	vertex to upper thigh	[^18^F]FDG
Ballal S [58]	2023	Eur J Nucl Med Mol Imaging	India	R	Radioiodine-resistant follicular thyroid cancers	restaging	30	PET/CT	[^68^Ga]Ga-DOTA.SA.FAPi	180 MBq	n.r.	vertex to upper thigh	[^18^F]FDG
Chen X [57]	2023	J Nucl Med	China	P	Lymphoma, various subtypes	diagnosis	7	PET/CT	[^68^Ga]Ga-FAPI	1.8–2.2 MBq/kg	60	skull base to upper thigh	[^18^F]FDG
Tian X [50]	2023	Jpn J Clin Oncol	China	n.r.	Unknown	staging	1	PET/CT	[^18^F]FAPI (nos.)	n.r.	n.r.	vertex to upper thigh	[^18^F]FDG
Dong A [51]	2023	Clin Nucl Med	China	P (trial)	Sarcomatoid renal cell carcinoma	staging	1	PET/CT	[^68^Ga]Ga-FAPI-04	n.r.	n.r.	vertex to upper thigh	NO
Tatar G [52]	2023	Mol Imaging Radionucl Ther	Turkey	n.r.	Papillary thyroid carcinoma	staging	1	PET/CT	[^68^Ga]Ga-FAPI-04	n.r.	n.r.	vertex to upper thigh	[^18^F]FDG
Ren JY [54]	2023	Clin Nucl Med	China	P (trial)	Colorectal cancer	restaging	1	PET/CT	[^18^F]FAPI-04	n.r.	n.r.	vertex to toes	[^18^F]FDG
Hoppner J [55]	2023	Sci Rep	Germany	R	Pancreatic ductal adenocarcinoma	restaging	7	PET/CT	[^68^Ga]Ga-FAPI-46	200–295 MBq	20 and 60 (dual time)	vertex to mid-thigh	NO
Zhang A [53]	2024	Eur Radiol	China	P	Primary extrapulmonary tumors in the chest	diagnosis, staging	n.r.	PET/CT	[^68^Ga]Ga-FAPI-04	2.5–3.5 MBq/kg	60 ± 10	skull base to upper femur	[^18^F]FDG
Has Simsek D [56]	2024	Eur J Nucl Med Mol Imaging	Turkey	P	NETs of various origin + paraganglioma + pheochromocytoma	restaging	10	PET/CT	[68Ga]Ga-FAPI-04	185–200 MBq	30–60	vertex to upper thigh	[^68^Ga]DOTATATE

Abbreviations: P—prospective, R—retrospective, FOV—field of view, GIST—gastrointestinal stromal tumor, NET—neuroendocrine tumor, NSCLC—non-small cell lung cancer, MALT—mucosa-associated lymphoid tissue, MEN—multiple endocrine neoplasia, CCR—colorectal cancer, n.r.—not reported.

**Table 3 ijms-25-07197-t003:** Comparison of FAPI vs. other tracer(s) in 19 studies on primary liver tumors.

First Author	FAPI Tracer	Other Tracer(s)	SUV FAPI vs. Other Tracer(s)	TBR FAPI vs. Other Tracer(s)	Advantages/Disadvantages
Li T [77]	[^68^Ga]Ga-FAPI (nos.)	[^18^F]FDG	Higher	Higher	[^68^Ga]Ga-FAPI reveals additional multiple liver lesions with variable FAPI uptake degree, not seen on [^18^F]FDG
Zhang J [78]	[^18^F]FAPI (nos.)	[^18^F]FDG	Higher	Higher	[^18^F]FAPI performed better than [^18^F]FDG in detecting HCC primary and lymph node lesions; peritoneal metastases better detected by [^18^F]FAPI
Li Z [76]	[^68^Ga]Ga-FAPI (nos.)	[^18^F]FDG	Higher	Higher	[^68^Ga]Ga-FAPI shows a remarkable TBR in patients with PEComa, 20 times higher than that of [^18^F]FDG
Suthar RR [72]	[^68^Ga]Ga-FAPI (nos.)	[^18^F]FDG	Higher	Higher	Superiority of [^68^Ga]Ga-FAPI over [^18^F]FDG in identifying fibrolamellar HCC (higher proportion of tumor-associated fibroblasts?)
Liu M [74]	[^68^Ga]Ga-FAPI (nos.)	[^18^F]FDG	Higher	n.r.	[^68^Ga]Ga-FAPI has potential in detecting hepatic adenocarcinoma; false positive findings in infectious conditions (pulmonary cryptococcosis)
Wu M [82]	[^68^Ga]Ga-FAPI-04	[^18^F]FDG	Comparable	n.r.	Volumetric indices on baseline [^68^Ga]Ga]-FAPI-04 were potentially prognostic factors to predict durable clinical benefit, PFS, and OS in unresectable HCC patients treated with combination of PD-1 and lenvatinib
Rajaraman V [81]	[^68^Ga]Ga-FAPI-04	[^18^F]FDG	Lower for HCC, higher for CC	Lower for HCC, higher for CC	[^68^Ga]Ga-FAPI-04] PET/CT clearly outperformed [^18^F]FDG in evaluating CC
Jinghua L [80]	[^68^Ga]Ga-DOTA-FAPI	[^18^F]FDG	Higher	Higher	[^68^Ga]Ga-DOTA-FAPI demonstrated higher uptake and sensitivity than [^18^F]FDG
Pabst KM [79]	[^68^Ga]Ga-FAPI-46	[^18^F]FDG	Higher	Higher (blood, liver)	[^68^Ga]Ga-FAPI-46 PET/CT displayed superior radiotracer uptake (especially Grade 3 tumors) and improved detection of lesions compared with [^18^F]FDG
Yao X [68]	[^68^Ga]Ga-FAPI (nos.)	[^18^F]FDG	Lower	n.r.	[^68^Ga]Ga-FAPI PET/MRI useful to detect invasion of blood vessels
Pan B [69]	Al^18^F-NOTA-FAPI-04	[^18^F]FDG	Higher	n.r.	Arc-shaped Al^18^F-NOTA-FAPI-04 uptake in one lesion and two additional lesions missed by [^18^F]FDG
Chen D [71]	[^18^F]-FAPI (nos.)	[^18^F]FDG	n.r.	n.r.	[^18^F]-FAPI better than [^18^F]FDG in detecting peritoneal metastases, even in early HCC; false positives due to post-treatment stromal fibrosis
Siripongsatian D [62]	[^68^Ga]Ga-FAPI-46	[^18^F]FDG	Higher	Higher	Higher uptake and TBR for [^68^Ga]Ga-FAPI-46 compared to [^18^F]FDG (better detection of hepatic lesions)
Pang Y [64]	[^68^Ga]Ga-FAPI (nos.)	[^18^F]FDG	Lower	Higher	[^68^Ga]Ga-FAPI PET/CT exhibited two times higher TBR than [^18^F]FDG
Ergül N [66]	[^68^Ga]GaDOTA-FAPI-04	[^18^F]FDG	Higher	n.r.	[^68^Ga]GaDOTA-FAPI-04 showed more intense uptake than [^18^F]FDG in peripheral regions of liver lesions (compared to central necrotic regions)
Siripongsatian D [63]	[^68^Ga]Ga-FAPI-46	[^18^F]FDG	n.r.	Higher	[^68^Ga]Ga-FAPI-46 PET/CT ensured change in patient management due to higher TBR in recurrent tumor and nodal metastases; some [^68^Ga]Ga-FAPI-46 active lesions not detected by [^18^F]FDG
Wang H [67]	[^68^Ga]Ga-FAPI-04	[^18^F]FDG	Comparable	Higher	[^68^Ga]Ga-FAPI-04 PET/CT demonstrated higher sensitivity than [^18^F]FDG in detecting intrahepatic HCC
Guo W [22]	[^68^Ga]Ga-FAPI-04	[^18^F]FDG	Higher	Higher	Sensitivity of [^68^Ga]Ga-FAPI-04 PET/CT in detecting primary hepatic tumors comparable to contrast-enhanced CT and liver MRI, but better than [^18^F]FDG
Shi X [21]	[^68^Ga]Ga-FAPI-04	[^18^F]FDG	Higher	Higher	Superiority of [^68^Ga]Ga-FAPI-04 in identifying primary malignancies compared with [^18^F]FDG

Abbreviations: SUV—standardized uptake value, TBR—tumor-to-background ratio, HCC—hepatocellular carcinoma, PFS—progression-free survival, OS—overall survival, nos.—not otherwise specified, n.r.—not reported.

**Table 4 ijms-25-07197-t004:** Comparison of FAPI vs. other tracer(s) in 35 studies on liver metastases.

First Author	Other Tracer(s)	SUV FAPI vs. Other Tracer(s)	TBR FAPI vs. Other Tracer(s)	Advantages/Disadvantages
Cheng Z [35]	[^18^F]FDG [^68^Ga]Ga-DOTATATE	Higher than [^18^F]FDG Lower than [^68^Ga]Ga-DOTATATE	n.r.	Outstanding performance of [^68^Ga]Ga-FAPI in detecting liver metastases from PNET, with remarkably high TBR. Small lymph node and skeletal metastases missed by both [^18^F]FDG and [^68^Ga]Ga-FAPI
Deng M [38]	[^18^F]FDG	n.r.	n.r.	[^68^Ga]Ga-FAPI better than [^18^F]FDG for metastatic liver lesions in gastrointestinal tumors
Giesel FL [34]	[^18^F]FDG	Higher	Higher	TBR of hepatic metastases to surrounding liver parenchyma significantly higher using [^68^Ga]Ga-FAPI than [^18^F]FDG (5.8 vs. 2.6; *p* = 0.011)
Kömek H [36]	[^18^F]FDG	Higher	Lower	Higher uptake of [^68^Ga]Ga-FAPI-04 than [^18^F]FDG in metastatic liver, lymph node, bone, and brain lesions, although < 1 cm sized
Kömek H [40]	[^68^Ga]Ga-DOTATATE	Lower	Higher	Better visualization of liver metastases from NET [^68^Ga]Ga-FAPI due to lower background physiological uptake in hepatic parenchyma, with respect to [^68^Ga]Ga-DOTATATE
Kreppel B [24]	[^18^F]FDG [^68^Ga]Ga-DOTATOC	Higher	n.r.	High correlation between [^68^Ga]Ga-DATA5m.SA.FAPi-positive tumor fraction with Ki-67 in liver metastases from NET (marker of aggressiveness and de-differentiation); non-concordant uptake pattern between [^18^F]FDG and [^68^Ga]Ga-DATA5m.SA.FAPi: [^18^F]FDG; uptake higher in the center of the lesion; [^68^Ga]Ga-DATA5m.SA.FAPi uptake more evident in the periphery
Kuyumcu S [33]	[^68^Ga]Ga-DOTATATE	n.r.	n.r.	Much more liver metastases evident on [^68^Ga]Ga-FAPI-04 PET/CT and missed by [^68^Ga]Ga-DOTATATE PET/CT
Pang Y [41]	[^18^F]FDG	Higher	n.r.	[^68^Ga]Ga-FAPI PET/CT higher than [^18^F]FDG in the primary lesion and in liver metastases, also with higher sensitivity
Sahin E [23]	[^18^F]FDG	n.r.	n.r.	Statistically higher number of liver lesions with [^68^Ga]Ga-DOTA-FAPI-PET/CT than with [^18^F]FDG-PET/CT
Wang L [29]	[^18^F]FDG	Lower	Higher	[^68^Ga]Ga-FAPI and [^18^F]FDG showed comparable performances in detecting liver, lung, and adrenal metastases
Barashki S [28]	[^68^Ga]Ga-DOTATATE [^131^I]mIBG	n.r.	n.r.	[^68^Ga]Ga-FAPI-46 PET/CT useful in showing progressive disease due to lung, liver, bone, and lymph nodes metastases
Can C [48]	[^18^F]FDG	Higher	Higher	[^68^Ga]Ga-FAPI better than [^18^F]FDG in correct disease TNM staging (change in patients’ overall disease status and management, although difference was statistically non-significant)
Elboga U [26]	[^18^F]FDG	Higher	n.r.	[^68^Ga]Ga-FAPI PET/CT useful as a complementary imaging modality in patients with inconclusive [^18^F]FDG for liver metastases; [^68^Ga]Ga-FAPI superior to [^18^F]FDG in detecting peritoneal involvement, with very high image quality
Erol Fenercioğlu O [32]	[^18^F]FDG	Lower	n.r.	[^18^F]FDG uptake in liver metastases from uveal melanoma higher than that of [^68^Ga]Ga-FAPI-4 PET/CT; for comparison, [^68^Ga]Ga-FAPI-4 uptake in osteoarthritis was higher than in liver metastases (unlike [^18^F]FDG)
Kou Y [30]	[^18^F]FDG	n.r.	n.r.	Liver metastases of MALT lymphoma visible only using FAPI PET/CT
Qin C [37]	[^18^F]FDG	Higher	n.r.	PET/MRI using [^68^Ga]Ga-FAPI showed more metastatic liver lesions, with higher uptake and a higher detection rate than [^18^F]FDG PET/CT
Tatar G [49]	[^18^F]FDG	Lower	n.r.	[^18^F]FDG better than [^68^Ga]Ga-FAPI in liver metastases from Kaposi sarcoma; it is, therefore, more suitable in determining disease extent and localizing all distant lesions
Wang H [39]	[^18^F]FDG [^11^C]acetate	n.r.	n.r.	Higher performance of [^68^Ga]Ga-FAPI-04 in detecting liver metastases of pancreatic NET, due to low background activity (false negative on both [^18^F]FDG and [^11^C]acetate); only [^68^Ga]Ga-FAPI-04 found the primitive tumor
Wu C [47]	[^18^F]FDG	n.r.	Higher	Higher detection rate and TBR in liver metastases from GISTs
Wu J [25]	[^18^F]FDG	Higher	Higher	[^68^Ga]Ga-FAPI exhibits significantly better diagnostic performances with respect to [^18^F]FDG in detecting liver (but also pleural, bone, and nodal) metastases from NSCLC
Xie F [46]	[^18^F]FDG	n.r.	n.r.	Single liver lesion detected by [^68^Ga]Ga-FAPI-04 thanks to the low background (low absolute uptake) compared to false negative [^18^F]FDG findings in renal cell carcinoma
Zhang Z [27]	[^18^F]FDG	Lower	n.r.	More liver metastases from pancreatic cancer detected using [^68^Ga]Ga-FAPI-04, thanks to the lower background than [^18^F]FDG (usually ring-shaped [^68^Ga]Ga-FAPI-04 uptake around the edge); absolute [^18^F]FDG SUVmax higher than that of [^68^Ga]Ga-FAPI-04, as per absolute value (unlike primary and lymph node lesions)
Li C [45]	[^18^F]FDG	Comparable	Higher	[^68^Ga]Ga-FAPI-04 superior to [^18^F]FDG in diagnosing liver metastasis from gastrointestinal cancers, thanks to higher TBR and TLR (lesions displayed more clearly)
Lin X [43]	[^18^F]FDG	Lower	Higher	[^68^Ga]Ga-FAPI-04 PET/CT improved tumor staging by detecting distant and, specifically, liver metastases from colorectal cancer (favorable TBR), especially in signet-ring/mucinous carcinoma, compared with [^18^F]FDG, thus prompting the optimization or adjustment of treatment decisions
Zhang Z [44]	[^18^F]FDG	Higher	n.r.	[^68^Ga]Ga-FAPI-04 PET/CT more sensitive than [^18^F]FDG for detecting metastatic liver lesions from GISTs
Ballal S [61]	[^18^F]FDG	n.r.	Higher	[^8^Ga]Ga-DOTA.SA.FAPi uptake was significantly higher in liver metastases than that of [^18^F]FDG
Liu Q [60]	[^18^F]FDG	Comparable	Higher	[^68^Ga]Ga-DOTA-FAPI-04 PET/CT detected more liver metastases than [^18^F]FDG, with a significantly higher tumor-to-liver background ratio of hepatic metastases
Dong Y [59]	[^18^F]FDG	Higher	Higher	Compared with [^18^F]FDG PET/CT, FAPI PET/CT showed more true positive liver lesions; they were more clearly delineated in 61.5% cases, equally in 30.8%, and inferiorly in 7.7%
Ballal S [58]	[^18^F]FDG	n.r	n.r.	[^68^Ga]Ga-DOTA.SA.FAPi had a higher detection rate for liver metastases compared to [^18^F]FDG
Tian X [50]	[^18^F]FDG	Higher	Higher	[^18^F]FAPI PET more useful in differentiating between focal fat-sparing and liver metastases in patients with fatty liver disease
Tatar G [52]	[^18^F]FDG	Higher	n.r.	[^68^Ga]Ga-FAPI-04 may add benefit in the restaging of metastatic DTC
Ren JY [54]	[^18^F]FDG	n.r.	n.r.	[^18^F]FAPI-04 performed similarly to [^18^F]FDG in detecting liver metastases from colorectal cancer; false positive [^18^F]FAPI-04 uptake in a vertebral hemangioma
Zhang A [53]	[^18^F]FDG	Comparable	Higher	[^68^Ga]Ga-FAPI has an overwhelming advantage on [^18^F]FDG in assessing metastases in different sites; tumor-to-liver ratio higher for [^68^Ga]Ga-FAPI than for [^18^F]FDG
Has Simsek D [56]	[^68^Ga]Ga-DOTATATE	Lower	n.r.	[^68^Ga]Ga-FAPI detected a lower number of liver metastases from NETs than [^68^Ga]Ga-DOTATATE (27/54 vs. 38/54), and the SUVmax of [^68^Ga]Ga-FAPI-positive lesions was much lower (median 5.1) than for [^68^Ga]Ga-DOTATATE-positive ones (median 16.6)

Abbreviations: SUV—standardized uptake value, TBR—tumor-to-background ratio, TLR—tumor-to-liver ratio, GIST—gastrointestinal stromal tumor, MALT—mucosa-associated lymphoid tissue, NSCLC—non-small cell lung cancer, NET—neuroendocrine tumor, n.r.—not reported.

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
