# Peer review of "Diagnostic Performances of PET/CT Using Fibroblast Activation Protein Inhibitors in Patients with Primary and Metastatic Liver Tumors: A Comprehensive Literature Review"

_ijms, 2024, doi:10.3390/ijms25137197_

Round 1
Reviewer 1 Report
Comments and Suggestions for Authors
In your literature review of the use of PET/CT in patients with primary and metastatic liver tumors you concluded that the use of FAPIs resulted in higher sensitivity in both primary and metastatic liver lesions than 18-FDG. The need for prospective research on wider populations is recommended.
In your medical centers can you demonstrate increases in survival rates particularly in primary liver tumors with the use of FAPI derived therapeutic agents?
Author Response
Reviewer's comment:
In your literature review of the use of PET/CT in patients with primary and metastatic liver tumors you concluded that the use of FAPIs resulted in higher sensitivity in both primary and metastatic liver lesions than 18-FDG. The need for prospective research on wider populations is recommended. In your medical centers can you demonstrate increases in survival rates particularly in primary liver tumors with the use of FAPI derived therapeutic agents?
Authors' response:
Dear Reviewer, thank for your observations. We definitely agree that prospective research on wider (and, possibly, more selected) populations is highly recommended in the future in order to clarify better the diagnostic and therapeutical strengths of FAPI derived agents. At the moment, we still do not use FAPI derived therapeutic agents for treating primary liver tumors in our practice, although new implementations are expected in our Center in the near future.
Reviewer 2 Report
Comments and Suggestions for Authors
The manuscript is a review which aims to evaluate the role of PET/CT using radiolabeled FAPIs in primary and/or metastatic liver lesions. The papers is of interest to readers, due to the prosiming results showed by FAPI-based radiotracers in various cancers. A main suggestion is to update the review to the beginning of 2024 at least, since it is now updated to January 2023 (one year and four months ago). Furthermore, the radiotracers' nomenclature should follow that suggested by the EANM Guidelines.
Comments on the Quality of English Language
Minor editing of English language required
Author Response
Reviewer's comment:
The manuscript is a review which aims to evaluate the role of PET/CT using radiolabeled FAPIs in primary and/or metastatic liver lesions. The papers is of interest to readers, due to the prosiming results showed by FAPI-based radiotracers in various cancers. A main suggestion is to update the review to the beginning of 2024 at least, since it is now updated to January 2023 (one year and four months ago).
Furthermore, the radiotracers' nomenclature should follow that suggested by the EANM Guidelines.
Authors' response:
Dear Reviewer, thank for your observations. As suggested, the literature search was updated to 1st January 2024: the number of screened papers was updated in accordance, new findings for more recent articles were reported in the manuscript text and the reference list was modified. Figure 1 (algorithm for articles retrieval and selection) and Tables 1-4 were updated accordingly. Changes in the revised manuscript (abstract and text) are highlighted in red.
Moreover, as suggested, the radiotracers' nomenclature was modified throughout the manuscript text following the more recent EANM Guidelines.
English language was also revised.
Reviewer 3 Report
Comments and Suggestions for Authors
This review gives a very good overview of radiolabled FAPIs used in primary and metastatic liver lesions compared with the conventional FDG tracer. We can see that FAPI tracers are superior to FDG tracer by providing higher sensitivity. The authors also acknowledge the lack of population size.
It will be great if the authors can also comment on the difference between FAPI tracers (i.e. 18F and 68Ga labelled ones)
Author Response
Reviewer's comment:
This review gives a very good overview of radiolabled FAPIs used in primary and metastatic liver lesions compared with the conventional FDG tracer. We can see that FAPI tracers are superior to FDG tracer by providing higher sensitivity. The authors also acknowledge the lack of population size.
It will be great if the authors can also comment on the difference between FAPI tracers (i.e. 18F and 68Ga labelled ones).
Authors' response:
Dear Reviewer, thank for your suggestions. Actually, the population size in most selected articles is small (also after updating the literature search), since 47 out of 76 papers included less than 15 patients and only 19 studied a larger population.
Thank you for pointing out the need for a comment on the difference between FAPI tracers used in the selected papers. This has been now added and highlighted in red in the revised manuscript at the end of the Discussion section.